# Metabolomic Profiling Reveals Distinct Signatures in Primary and Secondary Polycythemia

**DOI:** 10.3390/metabo15090630

**Published:** 2025-09-22

**Authors:** Murat Yıldırım, Batuhan Erdoğdu, Selim Sayın, Ozan Kaplan, Emine Koç, Mine Karadeniz, Bülent Karakaya, Mustafa Güney, Mustafa Çelebier, Meltem Aylı

**Affiliations:** 1Department of Hematology, Gülhane Training and Research Hospital, Ankara 06010, Türkiye; dr.batuhan@gmail.com (B.E.); drsayinselim@gmail.com (S.S.); drminekrdnz@gmail.com (M.K.); karakaya.b.23@gmail.com (B.K.); ayli.meltem@gmail.com (M.A.); 2Department of Analytical Chemistry, Faculty of Pharmacy, Hacettepe University, Ankara 06230, Türkiye; ozankaplan@hacettepe.edu.tr (O.K.); emikocc25@gmail.com (E.K.); celebier@hacettepe.edu.tr (M.Ç.); 3Blood Center, Gülhane Training and Research Hospital, Ankara 06010, Türkiye; mustafa.guney@sbu.edu.tr

**Keywords:** bioanalysis, biomarkers, metabolomics, LC/MS, polycythemia vera, secondary polycythemia

## Abstract

**Background/Objectives**: The differential diagnosis between primary polycythemia vera (PV) and secondary polycythemia (SP) presents significant clinical challenges owing to substantial phenotypic overlap. This investigation utilized untargeted metabolomic approaches to elucidate disease-specific metabolic perturbations and evaluate the metabolic consequences of cytoreductive therapeutic interventions. **Methods**: Plasma specimens obtained from PV patients (*n* = 40) and SP patients (*n* = 25) underwent comprehensive metabolomic profiling utilizing liquid chromatography–mass spectrometry (LC-MS) platforms. Multivariate statistical analyses, including principal component analysis (PCA), were employed in conjunction with pathway enrichment analyses to characterize disease-associated metabolic dysregulation. Additionally, receiving treatment (tPV) (*n* = 25) and not receiving treatment (ntPV) (*n* = 15) PV patients were compared to assess therapeutic metabolic effects. **Results**: Comprehensive metabolomic analysis identified 67 significantly altered metabolites between PV and SP patients, with 36 upregulated and 31 downregulated in PV. Key upregulated metabolites in PV included thyrotropin-releasing hormone, 3-sulfinoalanine, nicotinic acid adenine dinucleotide, and protoporphyrin IX, while 4-hydroxyretinoic acid and deoxyuridine were notably downregulated. Pathway enrichment analysis revealed disruptions in taurine, glutamate, nicotinate, and cysteine metabolism in PV. ntPV patients exhibited higher glucose and octanoyl-CoA levels compared to treated patients, indicating the normalization of glucose and fatty acid metabolism with cytoreductive therapy. ntPV was also associated with altered B-vitamin metabolism, including decreased nicotinic acid adenine dinucleotide and increased nicotinamide ribotide levels. Cross-comparison analysis revealed overlapping pathway enrichment in glutamate metabolism, nicotinate and nicotinamide metabolism, and cysteine metabolism between both comparisons. **Conclusions**: This study demonstrates that PV and SP exhibit fundamentally distinct metabolic signatures, providing novel insights into disease pathogenesis and potential diagnostic biomarkers. The identification of oxidative stress signatures, disrupted energy metabolism, and altered B-vitamin cofactor pathways distinguishes PV from SP at the molecular level. Cytoreductive therapy significantly normalizes metabolic dysregulation, particularly glucose and nucleotide metabolism, validating current therapeutic approaches while revealing broader systemic treatment effects. The metabolic signatures identified, particularly the combination of deoxyuridine, thyrotropin-releasing hormone, and oxidative stress metabolites, may serve as complementary diagnostic tools to traditional morphological and molecular approaches. These findings advance our understanding of myeloproliferative neoplasm pathophysiology and provide a foundation for developing metabolically targeted therapeutic strategies and precision medicine approaches in PV management.

## 1. Introduction

Polycythemia vera (PV) is a complex myeloproliferative neoplasm (MPN) characterized by the clonal and autonomous proliferation of hematopoietic stem cells, resulting in the pathological expansion of erythroid, myeloid, and megakaryocytic lineages. The hallmark feature of PV is increased erythropoiesis, resulting in elevated red blood cell mass, often accompanied by leukocytosis and thrombocytosis. The molecular pathogenesis of PV is predominantly driven by somatic mutations in the Janus kinase 2 (JAK2) gene. The JAK2 V617F point mutation is identified in approximately 95% of patients, whereas activating mutations in exon 12 of JAK2 account for most of the remaining cases [1]. These genetic alterations result in the constitutive activation of the JAK-STAT signaling pathway, promoting cytokine-independent hematopoietic proliferation and contributing to the clinical and laboratory manifestations of the disease.

Constitutive activation of the JAK-STAT signaling pathway resulting from these somatic mutations fundamentally alters cellular homeostasis and promotes aberrant hematopoietic proliferation. This molecular dysregulation is characterized by erythropoietin-independent erythroid colony formation and heightened sensitivity to hematopoietic growth factors, reflecting the loss of normal cytokine signaling pathways. The clinical manifestations of PV include thrombotic complications, microvascular disturbances, and the risk of progression to post-polycythemic myelofibrosis or secondary acute myeloid leukemia. Accordingly, early and accurate diagnosis of PV is critical, not only for risk stratification but also for timely therapeutic intervention to mitigate disease-related complications and improve long-term outcomes [1,2].

A persistent challenge in clinical practice is differentiating PV from secondary polycythemia (SP), a reactive condition arising from physiological or pathological elevations in erythropoietin (EPO) levels. SP may arise from chronic hypoxic states, such as chronic obstructive pulmonary disease, obstructive sleep apnea, and erythropoietin-secreting neoplasms. Despite the integration of serum erythropoietin measurements, bone marrow histopathology, and molecular testing for JAK2 mutations into current diagnostic algorithms, the substantial overlap in clinical features and laboratory findings between PV and SP often complicates accurate diagnosis [3]. While the majority of PV cases are straightforwardly diagnosed, the differentiation from SP can be challenging in a subset of patients, underscoring the need for more precise and discriminative biomarkers and approaches.

Recent advances in high-throughput molecular technologies have significantly enhanced our understanding of the mechanisms underlying the pathogenesis of various diseases. Metabolomics has emerged as a powerful approach for characterizing the complex biochemical alterations associated with various pathological states. Metabolomics provides valuable insights into both physiological processes and disease-associated biochemical alterations by enabling the comprehensive profiling of low-molecular-weight metabolites in biological samples [4,5]. This approach holds considerable promise for identifying novel diagnostic and prognostic biomarkers and uncovering previously unrecognized pathways implicated in hematologic malignancies and related disorders.

The application of metabolomics to hematological malignancies has revealed that metabolic reprogramming is a fundamental hallmark of disease progression. Malignant hematopoietic cells undergo profound alterations in their metabolic profiles, reflecting the increased energetic and biosynthetic demands associated with their uncontrolled proliferation. These shifts often manifest as distinct metabolic signatures that can serve as biomarkers of disease states and therapeutic responses. In the context of MPNs, previous studies have reported significant disruptions in central carbon metabolism, including enhanced glycolytic flux, increased glutaminolysis, and altered fatty acid oxidation pathways [6,7,8]. However, the specific metabolic landscape of PV, particularly in comparison to that of SP and healthy states, remains to be elucidated. A more detailed characterization of the PV-associated metabolome may not only yield novel diagnostic biomarkers capable of distinguishing PV from SP but also provide new insights into disease biology, uncovering potential metabolic vulnerabilities amenable to therapeutic targeting.

The dysregulation of cellular metabolism observed in PV extends beyond the metabolic demands of increased cellular proliferation and may play an active role in the pathogenesis and progression of the disease. Understanding these metabolic alterations may reveal novel therapeutic targets and prognostic indicators of PV. Recent evidence suggests that metabolic reprogramming within hematopoietic stem and progenitor cells significantly influences their self-renewal capacity, differentiation potential, and resistance to conventional therapies [9].

In this context, the present study aimed to elucidate the unique metabolic landscape of PV using comprehensive metabolomic profiling. By comparing metabolic signatures across PV and SP, we aimed to identify disease-specific metabolic patterns that may inform future diagnostic algorithms and guide targeted therapeutic interventions. This study represents an important step towards integrating metabolomic tools into the clinical management of PV and related myeloproliferative disorders.

## 2. Materials and Methods

### 2.1. Study Population

This prospective observational study was conducted between January 2024 and March 2025. The study included adult patients (≥18 years of age) who were diagnosed with PV or SP and were followed up at the hematology outpatient clinic. A total of 65 patients were included in the study, comprising 25 with SP and 40 with PV. A subgroup was conducted among patients with PV to compare those receiving treatment (tPV) (*n* = 25) and those not receiving treatment (ntPV) (*n* = 15). The diagnosis of PV was established based on the 2022 World Health Organization (WHO) criteria, including clinical, hematological, molecular, and histopathological findings. Among patients with PV, those receiving cytoreductive or disease-modifying treatment and those receiving phlebotomy and acetylsalicylic acid therapy were included in the study. It was also ensured that no phlebotomy procedures were performed within one month before sample collection. Also, bone marrow examination was performed for all PV patients.

PV patients were excluded if they had a history of other hematologic malignancies, active infections, pregnancy, hemoglobinopathy, hypertension, diabetes mellitus, hyperlipidemia, hypothyroidism, BMI > 30, severe hepatic or renal dysfunction, or any other significant comorbidities that could potentially confound the metabolomic analyses.

As secondary polycythemia encompasses a wide range of clinical entities that may markedly impact the metabolic profile, we confined our study to a well-defined and carefully selected subgroup of patients. The secondary polycythemia (control) group comprised individuals negative for both JAK2 V617F and JAK2 exon 12 mutations, with normal or slightly elevated serum EPO levels, hematocrit (Hct) values >50%, <65%. Bone marrow examination was performed for patients who have normal EPO but high hematocrit levels to exclude MPNs.

In the SP group, patient selection was guided by a comprehensive clinical and diagnostic evaluation, including arterial blood gas analysis (to identify methemoglobinemia and high-oxygen-affinity hemoglobinopathies via P_50_), capillary hemoglobin electrophoresis(variant hemoglobins), echocardiography, pulmonary function testing, whole-body contrast-enhanced computed tomography (in cases with suspected malignancy), and abdominal ultrasonography.

In addition to the standard exclusion criteria for the PV group, patients with clinically significant chronic obstructive pulmonary disease (COPD) or asthma requiring treatment, congenital or acquired cardiac shunts, known hemoglobinopathies (Hb D, thalassemia, Hb S, or high-oxygen-affinity variants), EPO-secreting tumors, sleep apnea, or obesity (BMI > 30), and thyroid disfunction were excluded.

Accordingly, the final SP cohort represented a clinically well-defined and highly selective group, consisting of patients with no identifiable etiology other than smoking, normal or slightly elevated erythropoietin concentrations, preserved or only mildly impaired pulmonary function, and absence of JAK2 V617F and JAK2 exon 12 mutations.

To reduce the influence of potential confounding variables on the metabolic profiles, patients in the PV and SP groups were matched primarily based on age, sex, and body mass index (BMI) values. Written informed consent was obtained from all participants prior to their enrollment in the study.

### 2.2. Sample Collection

Peripheral blood samples were collected into hemogram tubes containing EDTA following an overnight fast of 8–12 h. After collection, the samples were centrifuged at 3000 rpm for 10 min at +4 °C, and the plasma supernatant was carefully separated and transferred into clean tubes. These aliquots were stored at −80 °C until further analysis. On the day of analysis, all samples were transported to the laboratory using dry ice to maintain the cold chain and were analyzed.

### 2.3. Metabolite Extraction

Plasma samples were thawed at room temperature, and 0.1 mL of each sample was transferred into a microcentrifuge tube. Subsequently, 0.2 mL of cold methanol: water (9:1, *v*/*v*) was added. It was left at room temperature for 20 min. The mixture was vortexed for 1 min (IKA, VG 3,,Staufen, Germany), followed by protein precipitation through centrifugation at 12,000 rpm for 25 min at 4 °C (Hettich Universal 320 R, Tuttlingen, Germany). After centrifugation, 0.1 mL of the supernatant was transferred to a new microcentrifuge tube and evaporated to dryness using a vacuum centrifuge (Labconco, CentiVap, 7310030, Kansas City, MO, USA). The dried extract was reconstituted in 0.1 mL of acetonitrile: water (1:1, *v*/*v*), vortexed for 1 min, and centrifuged under the same conditions. Finally, 0.05 mL of the supernatant was transferred into LC/MS vials for subsequent analysis using quadrupole time-of-flight liquid chromatography/mass spectrometry (Q-TOF LC/MS). To ensure data quality and analytical consistency, pooled quality control (or QC) samples and extraction blanks were prepared for each experimental group. All samples underwent identical extraction procedures and were analyzed under uniform instrumental conditions.

### 2.4. LC/MS Analysis

Analyses were performed using a Q-TOF LC/MS system (Agilent Technologies 6530, Santa Clara, CA, USA). Sample injections (10 µL) were carried out using gradient elution on a reversed-phase chromatography column (2.1 × 100 mm, 2.5 µm; XBridge, Waters, Milford, MA, USA). The column was maintained at 35 °C, and the autosampler was set to 4 °C. The flow rate was 0.3 mL/min. The mobile phases consisted of (A) water with 0.1% formic acid and (B) acetonitrile with 0.1% formic acid. The elution gradient was as follows: 95% A at 0 min, 65% A at 2 min, 5% A at 8 min, returned to 95% A at 10 min, and maintained for 5 min as a post-run. The mass spectrometer operated in negative ionization mode, with a scan range of mass to charge (*m*/*z*) 75–1200. Sample injections were randomized, and QC samples and extraction blanks were injected every six runs to monitor analytical performance and ensure consistency.

### 2.5. Data Processing

Raw chromatographic data were exported in “.mzdata” format and processed using MZmine 2.53 software. Data processing included mass detection, chromatogram deconvolution, peak alignment, isotope grouping, and metabolite identification. Peaks present in extraction blank samples were excluded to eliminate background contaminants. To minimize noise, features with blank/QC signal intensity ratios exceeding 0.3 were removed. From the remaining peaks, those with percent relative standard deviation (%RSD) below 50% across quality control samples were retained to ensure analytical reproducibility. Peak areas were normalized using total peak area normalization. Each peak intensity was divided by the sum of all peak intensities within that sample, then multiplied by the average total intensity across all samples to maintain scale comparability. Metabolite identification followed a tiered approach according to the Metabolomics Standards Initiative (MSI) guidelines. Where possible, metabolites were confirmed at confidence Level 1 using our in-house library of authentic standards (Mass Spectrometry Metabolite Library, IROA Technologies, Virginia, USA) through retention time matching and MS/MS fragmentation pattern verification. For metabolites without available standards, putative identification was performed based on accurate mass matching (<5 ppm mass error) with metabolomic databases (HMDB, METLIN) and isotope pattern similarity. Principal component analysis (PCA) and partial least squares discriminant analysis (PLS-DA) were performed on normalized peak intensities. Univariate statistical analysis identified significantly altered metabolites between groups using Student’s *t*-test (*p* < 0.05) and fold change threshold (>1.5). Variable importance in projection (VIP) scores from PLS-DA models identified metabolites with the highest discriminatory power (VIP > 1.0). Pathway enrichment analysis was conducted on significantly altered metabolites using MetaboAnalyst 6.0 with the Kyoto Encyclopedia of Genes and Genomes (KEGG) pathway database. The hypergeometric test was applied for over-representation analysis, with pathway impact calculated using relative betweenness centrality. All statistical analyses, visualizations including PCA plots, heatmaps, and volcano plots were generated using MetaboAnalyst 6.0 software platform.”

### 2.6. Statistical Analysis

All statistical analyses were conducted using IBM SPSS software version 26.0. Continuous data were expressed as mean ± standard deviation, whereas categorical data were summarized as frequencies and percentages. The Kolmogorov–Smirnov test was employed to evaluate the normality of continuous variables, and Levene’s test was used to examine the homogeneity of variances. For variables with a normal distribution, group comparisons were made using Student’s *t*-test. In cases where the data were not normally distributed, the Mann–Whitney U test was utilized. Pearson’s chi-square test was applied for comparisons of categorical variables.

## 3. Results

### 3.1. Comparison Between Secondary Polycythemia and Polycythemia Vera Patients

A total of 65 patients were included in the study, comprising 25 with secondary polycythemia and 40 with polycythemia vera. The demographic and laboratory characteristics of both patient groups are shown in Table 1. There was no statistically significant difference between the two groups in terms of mean age (*p* = 0.659), sex distribution (*p* = 0.704), or white blood cell count (*p* = 0.337). However, platelet (Plt) counts were significantly higher, and hemoglobin (Hgb) levels were significantly lower, in the PV group compared to the SP group (*p* < 0.05), as detailed in Table 1.

### 3.2. Subgroup Analysis of Treated and Untreated Polycythemia Vera Patients

A subgroup analysis was conducted among patients with PV to compare those receiving treatment (tPV) (*n* = 25) and those not receiving treatment (ntPV) (*n* = 15). The detailed clinical and laboratory characteristics of the two subgroups are presented in Table 2. There were no significant differences between the groups in terms of PV duration (*p* = 0.476) or sex distribution (*p* = 0.191). However, the median age was significantly higher in the treatment group (*p* = 0.021). The prevalence of JAK2 mutations was similar between the two subgroups (*p* = 0.679). Cytoreductive therapy for PV is indicated for high-risk patients (age > 60 years or a history of thrombosis) and lower-risk patients who were intolerant to phlebotomy, have symptomatic or progressive splenomegaly, persistent leukocytosis, inadequate hematocrit control, or a high symptom burden. The therapeutic agents used in the treatment group included hydroxyurea, interferon, and ruxolitinib. Patients receiving treatment had significantly lower hemoglobin levels than ntPV patients (*p* = 0.030), whereas white blood cell and platelet counts did not differ significantly (*p* > 0.05). All comparative data are listed in Table 2.

### 3.3. Metabolomic Distinctions Between Disease Subgroups: PV vs. SP and tPV vs. ntPV

The PLS-DA plot in Figure 1 demonstrates a separation between the PV (red) and SP (green) groups, indicating distinct metabolic profiles in PV patients compared to SP individuals (Figure 1a). Figure 1b illustrates the significant metabolic differences between tPV (green) and ntPV (red), indicating distinct metabolic profiles between the treated and untreated subgroups.

Comprehensive metabolomic analysis identified 67 significantly altered metabolites between patients with PV and SP (*p* < 0.05, |fold change| > 1.5) (Appendix A). Of these, 36 metabolites were upregulated and 31 were downregulated in PV compared to those with SP. The most highly upregulated metabolites in PV included thyrotropin-releasing hormone (16.72-fold), 3a,7a,12a-trihydroxy-5b-cholestan-26-al (15.41-fold), 3-sulfinoalanine (13.14-fold), nicotinic acid adenine dinucleotide (8.25-fold), and protoporphyrin IX (7.42-fold). Additional significantly upregulated metabolites included cysteic acid (6.46-fold), campesterol (5.16-fold), D-glucose (4.29-fold), gamma-linolenoyl-CoA (3.46-fold), and inositol 1,3,4-trisphosphate (3.19-fold). The most substantially downregulated metabolites in PPV included phenol (−2.88-fold), deoxyuridine (−2.71-fold), 4-hydroxyretinoic acid (−2.62-fold), perillyl aldehyde (−2.48-fold), nudifloramide (−2.35-fold), and homovanillic acid (−2.35-fold).

A comparison between the ntPV and tPV groups identified 59 significantly altered metabolites (*p* < 0.05, |fold change| > 1.5), of which 34 were upregulated and 25 were downregulated in the ntPV group (Appendix A). The most substantially downregulated metabolites in ntPV patients included estrone glucuronide (−25.02-fold), phosphoribosyl formamidocarboxamide (−10.06-fold), D-glucose (−6.52-fold), nicotinic acid adenine dinucleotide (−6.37-fold), D-xylitol (−3.32-fold), L-3-hydroxykynurenine (−3.25-fold), and PS(16:0/16:0) (−3.09-fold). The most highly upregulated metabolites in ntPV patients included octanoyl-CoA (4.51-fold), dehydroepiandrosterone sulfate (4.21-fold), pyridoxal (4.03-fold), 3-(4-methylpent-3-en-1-yl)pent-2-enedioyl-CoA (3.60-fold), nicotinamide ribotide (3.39-fold), sinapyl alcohol (3.31-fold), and dUDP (2.96-fold), L-cysteine (2.87-fold).

Variable importance in projection (VIP) analysis identified deoxyuridine (VIP: 1.71), 4-hydroxyretinoic acid (1.52), 1-phosphatidyl-D-myo-inositol (1.47), nicotinic acid mononucleotide (1.45), nudifloramide (1.44), and hydrogen selenide (1.37) as the top discriminatory metabolites (Figure 2a). VIP analysis identified D-glucose (1.57), nicotinamide ribotide (1.56), hydrogen selenide (1.54), thiocyanate (1.48), L-3-hydroxykynurenine (1.45), inosinic acid (1.44), and phosphoribosyl formamidocarboxamide (1.40) as the most discriminatory metabolites between PV groups (Figure 2b).

Heatmap analysis of the top 30 differentially expressed metabolites revealed clear clustering patterns distinguishing PV (red) from SP (green) samples (Figure 3a), while the heatmap in Figure 3b illustrates the relative metabolite abundances across the ntPV (red) and tPV (green) groups.

Pathway enrichment analysis comparing PV and SP revealed significant alterations in taurine and hypotaurine (three hits, *p* = 0.014), glutamate (six hits, *p* = 0.0172), nicotinate and nicotinamide (five hits, *p* = 0.0172), cysteine metabolism (four hits, *p* = 0.0257), urea cycle (four hits, *p* = 0.033), and Warburg effect (six hits, *p* = 0.0374) (Figure 4a). Pathway enrichment analysis comparing tPV and ntPV showed significant alterations in glutamate (six hits, *p* = 0.0172), nicotinate and nicotinamide (five hits, *p* = 0.0172), cysteine (four hits, *p* = 0.0257), and purine metabolism (seven hits, *p* = 0.0392). Aspartate metabolism showed a trend towards significance (four hits, *p* = 0.0671) (Figure 4b).

### 3.4. Cross-Comparison Analysis

Both PV versus SP and tPV versus ntPV comparisons showed overlapping pathway enrichment in glutamate, nicotinate, and nicotinamide, and cysteine metabolism. The key metabolites contributing to these shared pathways included glutamic acid, succinic acid, L-aspartic acid, inosinic acid, deoxyuridine, and nicotinamide ribotide.

## 4. Discussion

The current metabolomic analysis revealed that PV and SP possess distinct metabolomic profiles characterized by 67 significantly altered metabolites that differentiate between the two conditions. These findings are consistent with emerging evidence suggesting that metabolic reprogramming is a fundamental characteristic of hematological malignancies that extends beyond the well-established genetic and cellular alterations [6,10]. Our results support the hypothesis proposed by He et al. (2024) that metabolomic profiling can offer complementary diagnostic information to traditional morphological and molecular approaches in MPNs [11].

The metabolic heterogeneity observed between PV and SP is consistent with findings from other malignancy comparisons, in which primary tumors consistently exhibit altered metabolic landscapes compared with reactive conditions [12]. This metabolic divergence likely reflects the intrinsic cellular reprogramming associated with clonal hematopoiesis, as opposed to the adaptive metabolic responses observed in patients with SP [13,14].

### 4.1. Metabolomic Differentiation Between SP and PV

#### 4.1.1. Central Carbon Metabolism and Energy Production

The 4.29-fold elevation of D-glucose in PV patients, alongside increased citric acid (1.91-fold) and succinic acid (2.34-fold), suggests active oxidative metabolism rather than the Warburg effect typically observed in hematological malignancies [15,16]. This aligns with recent findings indicating enhanced oxidative phosphorylation and mitochondrial metabolic activity in JAK2-mutated MPNs [17,18]. Studies by Rao et al. and He et al. further support this, demonstrating increased OXPHOS activity in JAK2-mutant hematopoietic stem and progenitor cells and elevated tricarboxylic acid (TCA) cycle intermediates in PV platelets, respectively [[11],].

This preserved TCA cycle activity in PV contrasts with the predominance of glycolysis in acute leukemias, potentially reflecting slower proliferation kinetics [19,20]. While earlier studies using functional imaging suggested normal glucose metabolism in MPNs, our direct metabolite quantification aligns with recent proteomic analyses showing upregulation of glycolytic enzymes in PV, particularly with JAK2V617F mutations.

#### 4.1.2. Amino Acid Metabolism and Cellular Signaling

The striking upregulation of 3-sulfinoalanine (13.14-fold) and cysteic acid (6.46-fold) in patients with PV compared to those with SP provides novel biochemical evidence of a distinct oxidative stress signature in PV, directly implicating the characteristic prothrombotic phenotype [21]. These metabolites, as direct products of cysteine oxidation, indicate enhanced sulfur amino acid catabolism and persistent redox imbalance specific to JAK2-driven myeloproliferation, extending beyond previous observations of increased reactive oxygen species (ROS) in PV neutrophils by identifying stable and quantifiable oxidation products [22,23,24]. The concurrent enrichment of cysteine and methionine metabolism suggests a broader dysregulation of sulfur amino acid pathways that may contribute to thrombotic risk through multiple mechanisms: direct endothelial damage, enhanced platelet hyperactivation, modulation of the coagulation cascade, and activation of the inflammatory-thrombotic interface [25,26,27,28]. These results distinguish PV from SP at the metabolic level, providing mechanistic insights into thrombotic complications and opening new avenues for redox-targeted interventions, biomarker development, and monitoring of therapeutic responses to JAK2 inhibitors.

Furthermore, perturbation of taurine and hypotaurine metabolism offers mechanistic insights into the altered cellular osmoregulation in patients with PV. This aligns with the findings in other cancers, including leukemia, where taurine dysregulation is associated with increased cellular stress and resistance to apoptosis [29,30,31]. A recent study demonstrated that bone marrow stromal cells actively support leukemogenesis by supplying taurine to leukemia stem cells via the taurine transporter SLC6A6 (TAUT). Taurine uptake promotes mTOR activation and glycolysis, facilitating cell survival and proliferation during oncogenic stress. Inhibition of either CDO1-driven stromal taurine synthesis or TAUT-mediated uptake significantly impairs leukemia progression and improves survival in preclinical acute myeloid leukemia models [32]. These findings are consistent with our observation that PV cells may exploit alternative osmolyte pathways, such as taurine/hypotaurine metabolism, to buffer oxidative stress and sustain cellular homeostasis in the context of JAK2-driven myeloproliferation.

#### 4.1.3. Nucleotide Metabolism and Cell Proliferation

In PV, the reduction in deoxyuridine (−2.71-fold) and inosine (−1.53-fold) appears paradoxical for a proliferative disease but reflects a metabolic reprogramming strategy seen in hematologic cancers. Evidence shows that malignant cells rewire their nucleotide metabolism to prefer the salvage pathway over de novo synthesis. For instance, Tran et al. (2024) demonstrated that circulating nucleosides like inosine are key to tumor purine pools, and inhibiting their salvage blocks tumor growth in vivo [33].

Similarly, broader reviews emphasize that successful malignancies depend on upregulated nucleoside transport and salvage enzyme activity to streamline the conversion of extracellular nucleosides into deoxyribonucleotides [34,35].

Thus, the low extracellular deoxyuridine and inosine in PV likely indicate increased cellular uptake and salvage activity in proliferating cells, rather than nucleotide substrate deficiency. This aligns with the known behavior of malignant hematopoietic and myeloid cells, which often overexpress nucleoside transporters (e.g., ENT1/2, CNTs) and salvage kinases to rapidly recycle nucleosides into DNA precursors [36].

Furthermore, PV patients exhibited a significant upregulation of inositol 1,3,4-trisphosphate (IP3) (3.19-fold), indicating disrupted phosphoinositide signaling. IP3, a crucial second messenger, mediates the release of intracellular calcium [37]. This finding corroborates prior observations of growth factor hypersensitivity in PV progenitors, particularly those with JAK2V617F mutations, even under JAK2 inhibition, suggesting alternative signaling mechanisms [38,39,40]. This links hyperactive IP3 signaling to PV pathogenesis, potentially contributing to abnormal calcium flux, megakaryocytic expansion, and thrombosis [41,42].

#### 4.1.4. Hormonal and Neuroendocrine Perturbations

Although thyroid dysfunction was an exclusion criterion in the study, the significant elevation of thyrotropin-releasing hormone (TRH) (16.72-fold) in PV patients is an unexpected finding, as thyroid dysfunction in hematological malignancies is typically attributed to autoimmune or iatrogenic causes rather than primary neuroendocrine dysregulation [43]. Given that TRH secretion is modulated by metabolic and inflammatory signals, this suggests that chronic inflammation and dysregulated pathways, such as those involving JAK2 mutations in PV, may alter neuroendocrine function [44,45,46]. This aligns with the understanding of cancer as a systemic metabolic disorder involving complex interactions among the immune, metabolic, and neuroendocrine systems [47,48].

Concurrently, the observed downregulation of estriol supports prior epidemiological data, indicating sex-related differences in PV presentation and outcomes [49]. These findings are consistent with those of studies demonstrating that steroid hormone metabolism can influence JAK-STAT signaling, thereby potentially modulating the disease severity in MPNs [50].

#### 4.1.5. Alterations in Redox-Active Metabolism and Antioxidant Defense

Elevated hydrogen selenide levels (VIP = 1.37) in patients with PV suggest altered selenium metabolism, which is crucial for the cellular antioxidant defense mechanism. Hydrogen selenide is vital for selenoprotein biosynthesis, including glutathione peroxidases (GPXs) and thioredoxin reductases, which maintain redox balance and prevent oxidative DNA damage.

Oxidative stress is a recognized hallmark of MPNs, with JAK2V617F mutations increasing ROS levels and contributing to genomic instability [51,52]. Selenium-dependent enzymes, particularly GPX1 and GPX4, mitigate the oxidative burden [53,54,55,56]. Therefore, increased hydrogen selenide levels in PV may represent both an adaptive response to oxidative stress and a potential contributor to PV pathogenesis via redox-sensitive regulation of intracellular signaling and DNA integrity.

#### 4.1.6. Impaired Retinoic Acid Metabolism and Megakaryocytic Dysfunction

Retinoic acid signaling is crucial for hematopoietic differentiation, with retinoid receptors (RARs and RXRs) regulating the expression of lineage-specific transcription factors. The megakaryocytic program is specifically controlled by transcription factors, such as GATA-1, FOG-1, and NFE2, which synergistically regulate megakaryocyte-specific gene expression [57,58,59,60]. Cross-talk between retinoic acid signaling and hematopoietic transcription factors has been observed in erythropoietic differentiation, where retinoic acid, hypoxia, and GATA factors cooperatively regulate gene expression [60,61]

Our metabolomic analysis identified a significant downregulation of 4-hydroxyretinoic acid in PV patients compared to SP (−2.62-fold), offering new insight into impaired retinoid signaling in PV. This supports the hypothesis that disrupted retinoid metabolism contributes to characteristic features like thrombocytosis and megakaryocytic hyperplasia. Since RA derivatives inhibit JAK2, STAT3, and STAT5 phosphorylation [62], the depletion of 4-hydroxyretinoic acid may allow PV cells to evade growth inhibition. This reduction could enhance megakaryocytic proliferation by impairing the regulation of the GATA-1/FOG-1/NFE2 transcriptional network, which controls megakaryocyte maturation and platelet production [63].

### 4.2. Impact of Cytoreductive Therapy on Metabolic Profiles

#### 4.2.1. Therapeutic Normalization of Glucose Metabolism

Our observation of restored glucose metabolism in patients with PV following cytoreductive therapy represents a novel and potentially significant systemic treatment effect. While previous studies have highlighted the cardiovascular benefits of cytoreductive therapy, particularly hydroxyurea, in reducing thrombosis and inflammation in PV, its metabolic implications have largely remained unexplored [64,65].

This study offers the first clinical metabolomic-based evidence that cytoreductive therapy can normalize dysregulated glucose metabolism in hematologic malignancies. This is highly relevant, as JAK2-driven PV models demonstrated profound metabolic dysregulation, including hypoglycemia and cachexia, supporting the concept that these malignancies exhibit metabolic alterations that may be therapeutically targeted [66].

This finding contrasts with the patterns observed in solid tumors, where cytoreductive or cytotoxic therapies often fail to reverse cancer-associated metabolic reprogramming [67,68]. One potential explanation for this discrepancy lies in the differences in the tumor microenvironment and cellular plasticity between hematologic and solid malignancies. PV tumor clones arise in a relatively permissive environment, where constitutive JAK2 signaling drives glycolysis, glucose uptake, and proliferation but remains responsive to systemic metabolic modulation [69]. Furthermore, the recognized roles of chronic inflammation and insulin resistance in PV suggest that cytoreductive therapy may indirectly improve glucose metabolism by reducing the inflammatory burden and cytokine-driven insulin signaling disruptions [70].

#### 4.2.2. B-Vitamin Metabolism and Co-Factor Availability

A significant downregulation of nicotinic acid adenine dinucleotide (NAAD) (−6.37-fold) and upregulation of nicotinamide ribotide (3.39-fold) in untreated PV patients indicated dysregulated niacin (vitamin B3) metabolism and NAD+ biosynthesis. Reduced NAAD suggests impaired de novo NAD+ synthesis through the kynurenine pathway, further supported by decreased L-3-hydroxykynurenine (−3.25-fold), a key intermediate in tryptophan-to-NAD+ conversion [71].

NAD+ is a vital cofactor for energy metabolism (glycolysis, fatty acid oxidation, TCA cycle) [72,73], and its disruption may contribute to the energetic imbalance in MPNs. The elevated nicotinamide ribotide in treated patients likely reflects a compensatory salvage pathway activation to maintain NAD+ pools. This suggests that untreated patients exhibit early metabolic stress responses to preserve NAD+-dependent functions.

The substantial downregulation of phosphoribosyl formamidocarboxamide (−10.06-fold, VIP = 1.40) suggests impaired folate-dependent one-carbon metabolism, which is essential for purine synthesis and DNA methylation [74].

This finding, combined with altered B-vitamin cofactor availability, indicates widespread metabolic reprogramming that affects nucleotide biosynthesis and epigenetic regulation in PV pathogenesis. These B-vitamin metabolic alterations may serve as early biomarkers for disease progression and suggest potential therapeutic targets for early-stage PV management through targeted vitamin supplementation strategies.

Our data point to a significant dysregulation in the NAD+ biosynthetic pathway, a key component of B-vitamin metabolism, particularly in untreated PV patients. However, we contend that it is premature to recommend supplementation trials; instead, future studies should focus on mechanistic research to determine if this pathway represents a viable therapeutic target.

#### 4.2.3. Lipid Metabolism and Cellular Membrane Composition

The elevation of octanoyl-CoA (4.51-fold) in ntPV patients provides evidence of altered fatty acid metabolism in PV, extending previous observations of dyslipidemia in MPNs [75]. The normalization of fatty acid metabolites following treatment supports the comprehensive metabolic benefits of cytoreductive therapy.

### 4.3. Shared Metabolic Pathways and Disease Mechanisms

#### 4.3.1. Glutamate Metabolism

The consistent enrichment of glutamate metabolism across multiple comparisons identified this pathway as a central metabolic node in the pathogenesis of PV. These findings support recent studies demonstrating that glutamine addiction represents a metabolic vulnerability in hematological malignancies [76]. The glutamate metabolic signature identified in the present study aligns with the observations of Coen et al. (2023), who demonstrated enhanced glutaminolysis in JAK2-mutated cells [77].

The centrality of glutamate metabolism in our analysis is consistent with its role in maintaining cellular redox balance through glutathione synthesis, which is particularly relevant given the oxidative stress signature observed. This metabolic phenotype differs from that of solid tumors, where glutamine metabolism primarily supports biomass production rather than antioxidant defense [78].

#### 4.3.2. Purine Metabolism and DNA Synthesis

The enrichment of purine metabolism (*p* = 0.039) in the treatment comparison provides evidence of altered nucleotide homeostasis in PV, consistent with previous studies demonstrating genomic instability in MPNs [79]. These findings extend the work of Putter et al. (2021), who identified molecular genetic alterations in PV patients, including JAK2 mutations and their impact on cellular signaling pathways, but did not examine the downstream metabolic consequences of these genetic defects [80]. Although PV has been characterized as having relatively genome-stable conditions compared to other MPNs [81], the JAK2 V617F mutation present in nearly 98% of PV cases may still contribute to DNA damage response alterations [82], potentially linking the observed purine metabolism dysregulation to underlying genomic stress mechanisms.

### 4.4. Clinical Implications and Diagnostic Potential

#### 4.4.1. Biomarker Development

The distinct metabolic signatures identified in this study provide a foundation for developing clinically applicable diagnostic biomarkers. The Variable Importance in Projection analysis identified specific metabolites with high discriminatory power, including deoxyuridine (VIP: 1.71) and IP3 (VIP: 1.28), which could be incorporated into targeted metabolomic panels. These findings complement recent studies developing metabolomic diagnostic approaches for other hematological malignancies [83,84,85]

The diagnostic potential of metabolomic profiling is particularly relevant, given the ongoing challenges in distinguishing PV from secondary causes of polycythemia. Our metabolic signature approach may provide complementary information to existing molecular diagnostic methods, potentially improving diagnostic accuracy and reducing the time to diagnosis.

#### 4.4.2. Therapeutic Target Identification

The metabolic perturbations identified in this study reveal potential therapeutic vulnerabilities that extend beyond the traditional cytoreductive approaches. The oxidative stress signature suggests that antioxidant interventions may provide therapeutic benefits, consistent with recent clinical trials examining antioxidant supplementation in MPNs. Similarly, B-vitamin pathway perturbations indicate potential therapeutic opportunities for cofactor supplementation.

The metabolic normalization observed following cytoreductive therapy validates current therapeutic approaches and provides mechanistic insights into the benefits of treatment. These findings support early therapeutic interventions to prevent metabolic complications, consistent with current clinical guidelines emphasizing prompt treatment initiation.

### 4.5. Study Limitations

Several limitations should be acknowledged when interpreting these results. The cross-sectional study design limits the assessment of temporal metabolic changes and their relationship with disease progression. Additionally, the heterogeneity in treatment regimens among treated patients may have introduced confounding variables affecting metabolic profiles.

Notably, the stringent inclusion and exclusion criteria for the SP cohort, while necessary to create a metabolically uniform comparison group, may limit the generalizability of our findings and may not capture the full metabolic diversity of SP. This approach intentionally reduced the heterogeneity typical of SP cases seen in clinical practice. Therefore, the metabolic distinctions identified may be most applicable to a specific subset of the SP population rather than its entire spectrum. Future studies incorporating more heterogeneous SP etiologies will be essential to validate whether the identified metabolic signatures consistently distinguish PV from various SP subtypes.

Second, in our subgroup analysis, the treated PV patients were significantly older than the untreated group. This age imbalance is an inherent consequence of current treatment guidelines, which often recommend cytoreductive therapy for older patients. We did not adjust for age due to sample size, so age-related metabolic changes are a potential confounder. These findings require cautious interpretation pending future studies with age-matched or statistically adjusted cohorts.

Polycytemia without an identifiable etiology, evaluation of erythrocyte 2,3-DPG levels, bisphosphoglycerate mutase deficiency, and oxygen-sensing pathway gene mutations (EpoR, VHL, PHD2) were not possible due to technical limitations.

Metabolite identification represents another important limitation; while a subset of metabolites was confirmed at MSI Level 1 using authentic standards and MS/MS fragmentation, many metabolites remain at Level 2 (putative annotation based on accurate mass and database matching). Although not all metabolites achieved Level 1 confirmation, the consistency of pathway enrichment patterns and the validity of relative quantification support the biological relevance of the observed metabolic differences between PV and SP. Overall, we view this study as hypothesis-generating rather than hypothesis-testing. The identified metabolic signatures require validation in independent cohorts with appropriate multiple testing corrections, and such confirmatory efforts will be essential before any clinical application.

## 5. Conclusions

This comprehensive metabolomic analysis revealed that PV and SP exhibit fundamentally different metabolic landscapes, reflecting distinct disease mechanisms. The identification of specific metabolic signatures provides new insights into PV pathogenesis and offers potential avenues for improved diagnosis and therapeutic intervention. The normalization of metabolic profiles following cytoreductive therapy validates current treatment approaches, revealing the broader systemic effects of therapeutic interventions.

These findings advance our understanding of the metabolic basis of MPN and provide a foundation for developing metabolically targeted therapeutic strategies. The integration of metabolomics into clinical practice may ultimately improve outcomes for patients with polycythemia through enhanced diagnostic accuracy, risk stratification, and personalized treatment approaches.

## Figures and Tables

**Figure 1 metabolites-15-00630-f001:**
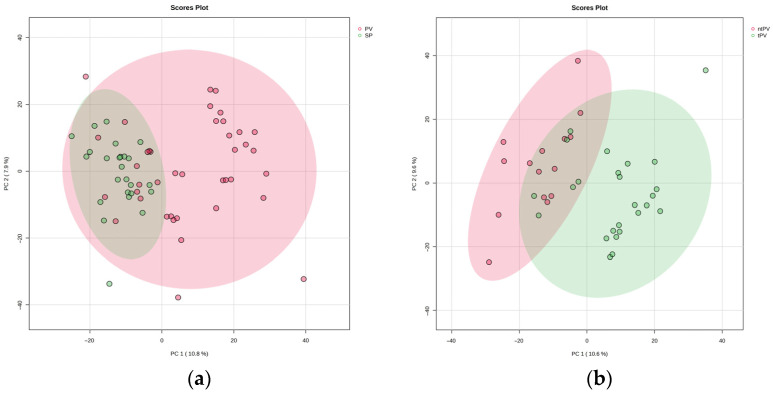
A PCA plot showing the separation of metabolomic profiles across (**a**) the Polycythemia Vera group (PV, red), Secondary Polycythemia group (SP, green), and (**b**) Polycythemia Vera patients receiving treatment (tPV, green) and Polycythemia Vera patients not receiving treatment (ntPV, red).

**Figure 2 metabolites-15-00630-f002:**
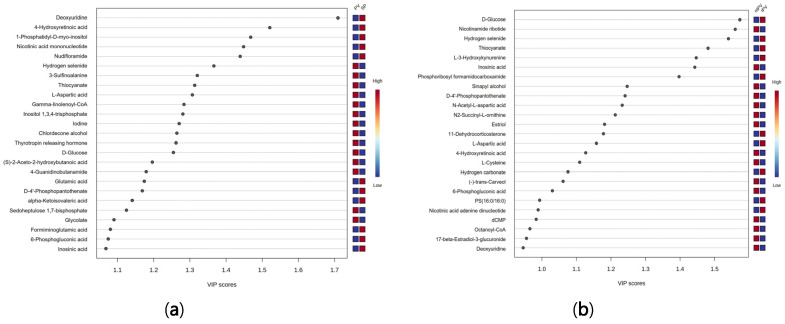
VIP score plots illustrating the key metabolites that most significantly distinguish between (**a**) SP and PV and (**b**) ntPV and tPV groups. Metabolites with VIP scores exceeding 1.5 were considered to have significant contributions.

**Figure 3 metabolites-15-00630-f003:**
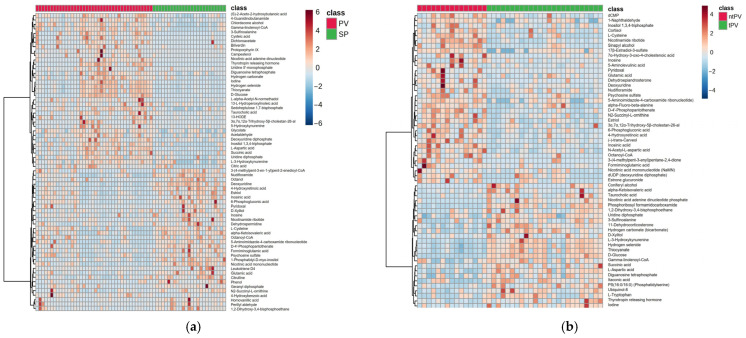
Heatmap analysis showed distinct clustering patterns between (**a**) PV (red) and SP (green), (**b**) the ntPV (red) and tPV (green) groups.

**Figure 4 metabolites-15-00630-f004:**
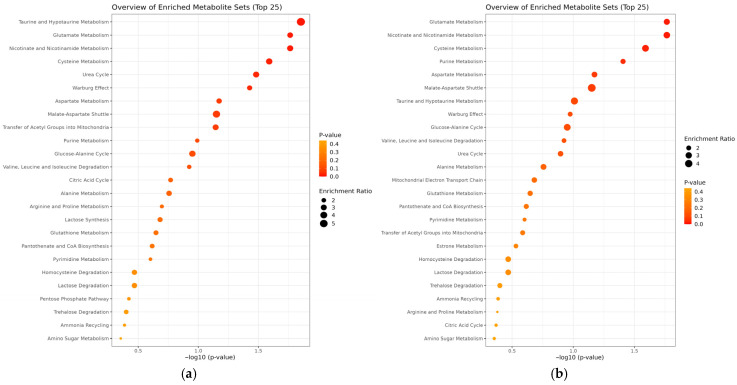
Pathway enrichment analysis comparing (**a**) the SP and PV groups and (**b**) the tPV and ntPV groups. The plot highlights the most significantly enriched metabolic pathways, with larger circles representing a higher pathway impact and darker colors indicating lower *p*-values (greater statistical significance).

**Table 1 metabolites-15-00630-t001:** Demographic characteristics and laboratory data of the PV and SP groups.

	SP (*n* = 25)	PV (*n* = 40)	*p*-Value
Mean Age (years) ± SD	45.8 (±13.6)	54.3 (±13.1)	0.659
Gender (*n*) (%)			
Male	18 (72%)	27 (67.5%)	0.704
Female	7 (28%)	13 (32.5%)	
Mean WBC (mm^3^) ± SD	9222 (±2507.5.1)	8232.8 (±2653.3)	0.337
Mean Hgb (g/dL) ± SD	16.9 (±1.3)	14.7 (±2.7)	0.00
Mean Plt (mm^3^) ± SD	257,560 (±73,158.5)	365,907.5 (±170,619.6)	0.00

SP: Secondary Polycythemia, PV: Polycythemia Vera, WBC: White Blood Cell, Hgb: Hemoglobin, Plt: Platelet, SD: Standard Deviation.

**Table 2 metabolites-15-00630-t002:** Comparison of demographic characteristics and laboratory parameters between tPV and ntPV patients.

	tPV (*n* = 25)	ntPV (*n* = 15)	*p*-Value
Mean Age (years) ± SD	58.2 (±13.7)	47.6 (±9.0)	0.021
Gender (*n*) (%)			
Male	15 (60%)	12 (80%)	0.191
Female	10 (40%)	3 (20%)	
Mean Duration of PV (year) ± SD	4.8 ± 4.0	4.8 ± 5.8	0.476
Mean WBC (mm^3^) ± SD	7487.7 (±2811.0)	9474.7 (±1849.6)	0.528
Mean Hgb (g/dL) ± SD	14.1 (±3.1)	15.6 (±1.1)	0.030
Mean Plt (mm^3^) ± SD	320,640 (±14,762)	441,353.3 (±184,326)	0.384
Genetic Mutation Status (%)			0.679
JAK-2 V617F mutation	24 (96%)	15 (100%)	
JAK-2 Exon 12 mutation	1 (4%)	-	
Therapeutic Agent (%)			
Hydroxyurea	21 (84%)		
Interferon	2 (8%)		
Ruxolitinib	2 (8%)		
Treatment Indications (%)			
Age > 60	7 (28%)		
History of thrombosis	9 (36%)		
Persistent leukocytosis, inadequate hematocrit control,	4 (16%)		
Intolerance to phlebotomy	4 (16%)		

PV: Polycythemia Vera patients receiving treatment, ntPV: Polycythemia Vera patients not receiving treatment, WBC: White Blood Cell, Hgb: Hemoglobin, Plt: Platelet, SD: Standard Deviation.

## Data Availability

The data that support the findings of this study are not publicly available due to privacy reasons but are available from the corresponding author upon reasonable request.

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
