# Peer review of "Metabolomic Profiling Reveals Distinct Signatures in Primary and Secondary Polycythemia"

_metabolites, 2025, doi:10.3390/metabo15090630_

Round 1

Reviewer 1 Report

Comments and Suggestions for Authors

The study is timely and relevant, but in its current form, the manuscript requires major revision before it can be considered for publication:

1. The inclusion and exclusion criteria for secondary polycythemia patients are highly restrictive. Does this narrow cohort accurately reflect the heterogeneity of SP cases seen in clinical practice? Could this limit the generalizability of the findings?
2. How was the sample size determined? Was a power calculation performed to justify 40 PV and 25 SP patients?
3. In the subgroup analysis of treated versus untreated PV patients, the treated group was significantly older. Could age-related metabolic changes confound the observed differences? Have the authors adjusted for this?
4. Putative identification was used for metabolite annotation. Were any metabolites confirmed with authentic standards or MS/MS fragmentation? Without this, how reliable are the reported fold changes?
5. The normalization method and statistical thresholds (fold change > 1.5, p < 0.05) may be prone to false positives. Were multiple testing corrections (e.g., FDR) applied?
6. The role of deoxyuridine depletion as a biomarker is highlighted. Could this observation be confounded by dietary or pharmacological factors not reported in the study?
7. The elevation of thyrotropin-releasing hormone in PV is unexpected. Were thyroid hormone levels measured in these patients to exclude subtle dysfunction?
8. How do the findings on nicotinamide and B-vitamin metabolism relate to potential clinical interventions? Would the authors suggest vitamin supplementation studies, or is this premature?

Author Response

We thank the reviewer for their constructive feedback on our manuscript. We agree with the reviewer's suggestions for improvement and have addressed them as follows:

Comment 1.1: The inclusion and exclusion criteria for secondary polycythemia patients are highly restrictive. Does this narrow cohort accurately reflect the heterogeneity of SP cases seen in clinical practice? Could this limit the generalizability of the findings?.

Response 1.1: Secondary polycythemia encompasses a wide and heterogeneous spectrum of etiologies, ranging from hypoxic lung diseases and congenital heart defects to high-oxygen-affinity hemoglobinopathies and EPO-secreting tumors.

In our metabolomics study, we intentionally applied strict inclusion and exclusion criteria to minimize the influence of variables that could significantly alter metabolic profiles (e.g., severe COPD, sleep apnea, morbid obesity, hemoglobinopathies, malignant causes of erythrocytosis, steroid use). However, this narrow patient group still accounts for 30–40% of the CP cases we encounter in our daily practice.

Our goal was to create a metabolically "clean" comparison group in which differences could be more reliably attributed to the underlying biology of polycythemia vera rather than to various comorbidities.

We acknowledge that this approach reduces the heterogeneity of the CP group and therefore limits the generalizability of our findings to the broader CP population.

However, we view this study as a first step toward defining the fundamental metabolomic distinctions between PV and SP.Future studies with larger and more diverse SP cohorts, including patients with hypoxia-induced and tumor-associated forms, will be essential to confirm and expand our findings.

We have clarified this limitation in the revised manuscript (see “Study Limitations” section) and highlighted the need for broader cohort studies as a direction for future research.

“Our study has several limitations. Notably, the stringent inclusion and exclusion criteria for the secondary polycythemia cohort, while necessary to create a metabolically uniform comparison group, may limit the generalizability of our findings. This approach intentionally reduced the heterogeneity typical of secondary polycythemia cases seen in broader clinical practice. Therefore, the metabolic distinctions identified may be most applicable to a specific subset of the SP population rather than its entire spectrum”

Comment 1.2: How was the sample size determined? Was a power calculation performed to justify 40 PV and 25 SP patients?
Response 1.2: In this study, the sample size was determined by power analysis to be 45 patients with PV and 45 patients with SP (Figure 1).

Initially, we could easily reach this number by including all secondary polycythemia patients. However, the strict inclusion and exclusion criteria we implemented to reduce heterogeneity in the SP group, significantly reduced our sample size. Patients were more satisfied with rapid symptomatic relief through phlebotomy, so they were unwilling or unable to undergo comprehensive diagnostic evaluations to rule out alternative causes of erythrocytosis (such as arterial blood gas testing and CT imaging). Furthermore, the cohort size was further reduced by excluding patients with chronic obstructive pulmonary disease (one of the most common SP subgroups) requiring inhaled or systemic therapy. Steroid use was a factor that could definitely affect metabolism that we could not ignore.

All these factors mentioned above caused the SP group to be smaller than the PV group.

Importantly, our final sample size (40 PV and 25 SP patients) is comparable to or larger than several published metabolomics studies in hematological disorders, where discovery cohorts typically range from 15 to 30 patients per group.

The robustness of our findings is further supported by the clear separation of groups in multivariate analyses (PCA and PLS-DA), the identification of multiple metabolites with high discriminatory power, and the consistency of pathway enrichment results.

We fully acknowledge that a larger cohort with prospective power calculations would have strengthened the generalizability of our findings. This was noted as a limitation in the revised article, and we emphasize that future validation in larger, more heterogeneous cohorts will be important.

Comment 1.3: In the subgroup analysis of treated versus untreated PV patients, the treated group was significantly older. Could age-related metabolic changes confound the observed differences? Have the authors adjusted for this?

Response 1.3: It is correct that the treated PV (tPV) group was significantly older than the untreated PV (ntPV) group, and we acknowledge that age-related metabolic changes could represent a potential confounder.

The most important reason for this imbalance is that current MPN guidelines recommend cytoreductive treatment for patients over 65 years of age and for those at high risk of thromboembolic events, whereas younger, lower-risk patients are generally managed with phlebotomy and acetylsalicylic acid alone. Thus, treatment status and age are inherently linked in clinical practice.

Although we did not perform formal multivariate adjustment for age due to the limited sample size, we believe the observed metabolic differences are unlikely to be explained by age alone, as they align with known PV-related metabolic perturbations and show consistency with prior mechanistic studies.

We have now explicitly acknowledged this as a limitation in the revised manuscript and emphasized that future larger studies with age-matched or statistically adjusted cohorts will be required to confirm these findings.

We have clarified this limitation in the revised manuscript

‘’Second, in our subgroup analysis, the treated polycythemia vera (PV) patients were significantly older than the untreated group. This age imbalance is an inherent consequence of current treatment guidelines, which often recommend cytoreductive therapy for older patients. We did not adjust for age due to sample size, so age-related metabolic changes are a potential confounder. These findings require cautious interpretation pending future studies with age-matched or statistically adjusted cohorts.’’

Comment 1.4:Putative identification was used for metabolite annotation. Were any metabolites confirmed with authentic standards or MS/MS fragmentation? Without this, how reliable are the reported fold changes?

Response 1.4: In this exploratory study, we employed a tiered metabolite identification approach. A subset of key metabolites was confirmed at MSI Level 1 using our in-house library of authentic standards (Mass Spectrometry Metabolite Library from IROA Technologies), with retention time matching and MS/MS fragmentation pattern verification. This included critical metabolites such as glucose, amino acids, and several nucleotides that showed significant alterations between groups.

For metabolites without available standards in our library, putative identification (MSI Level 2) was performed using accurate mass matching (<5 ppm error), isotope pattern analysis, and database comparison against HMDB and METLIN in MZmine 2.53.

Regarding fold change reliability: The reported fold changes are based on integrated peak areas from consistently detected chromatographic peaks across all samples, normalized using total peak area normalization. This relative quantification approach remains valid regardless of identification level, as it depends on consistent peak detection and integration rather than absolute compound identity. The same chromatographic peak is tracked across all samples, ensuring that fold change calculations accurately reflect relative abundance differences between groups.

We implemented several quality control measures to ensure data reliability:

(a) Stringent peak filtering with blank subtraction (features with blank/QC ratio >0.3 removed)

(b) Retention of peaks with %RSD <50% across samples to ensure analytical reproducibility

(c) Statistical filtering using fold change >1.5 and p < 0.05 thresholds

(d) Biological validation through pathway enrichment analysis showing coherent metabolic alterations

We acknowledge that the mixture of Level 1 confirmed and putative identifications represents a limitation. However, the consistency of our pathway findings, the biological plausibility of the results, and the large effect sizes observed (fold changes ranging from 2.7 to 16.7 for key metabolites) support the validity of our conclusions.

Revised Method section;

“2.5. Data Processing

Raw chromatographic data were exported in ".mzdata" format and processed using MZmine 2.53 software. Data processing included mass detection, chromatogram deconvolution, peak alignment, isotope grouping, and metabolite identification. Peaks present in extraction blank samples were excluded to eliminate background contaminants. To minimize noise, features with blank/QC signal intensity ratios exceeding 0.3 were removed. From the remaining peaks, those with percent relative standard deviation (%RSD) below 50% across quality control samples were retained to ensure analytical reproducibility. Peak areas were normalized using total peak area normalization. Each peak intensity was divided by the sum of all peak intensities within that sample, then multiplied by the average total intensity across all samples to maintain scale comparability. Metabolite identification followed a tiered approach according to the Metabolomics Standards Initiative (MSI) guidelines. Where possible, metabolites were confirmed at confidence Level 1 using our in-house library of authentic standards (Mass Spectrometry Metabolite Library, IROA Technologies, Virginia, USA) through retention time matching and MS/MS fragmentation pattern verification. For metabolites without available standards, putative identification was performed based on accurate mass matching (<5 ppm mass error) with metabolomics databases (HMDB, METLIN) and isotope pattern similarity. Principal component analysis (PCA) and partial least squares discriminant analysis (PLS-DA) were performed on normalized peak intensities. Univariate statistical analysis identified significantly altered metabolites between groups using Student's t-test (p < 0.05) and fold change threshold (>1.5). Variable importance in projection (VIP) scores from PLS-DA models identified metabolites with highest discriminatory power (VIP > 1.0). Pathway enrichment analysis was conducted on significantly altered metabolites using MetaboAnalyst 6.0 with the Kyoto Encyclopedia of Genes and Genomes (KEGG) pathway database. Hypergeometric test was applied for over-representation analysis, with pathway impact calculated using relative betweenness centrality. All statistical analyses, visualizations including PCA plots, heatmaps, and volcano plots were generated using MetaboAnalyst 6.0 software platform.”

Comment 1.5: The normalization method and statistical thresholds (fold change > 1.5, p < 0.05) may be prone to false positives. Were multiple testing corrections (e.g., FDR) applied?

Response 1.5: We thank the reviewer for this insightful comment, which rightly highlights a critical consideration in high-dimensional metabolomic data analysis. We acknowledge that multiple testing correction was not applied in this exploratory metabolomics study. The thresholds used (fold change >1.5, p <0.05) are standard in untargeted metabolomics discovery studies, though we recognize this may increase the risk of false positives.  However, several factors support the robustness of our findings despite the absence of FDR correction:

  1. Large effect sizes: Many of our key metabolites showed substantial fold changes (2.7-16.7), well above the 1.5 threshold. Such large effect sizes are less likely to be false positives compared to marginal changes.
  2. Biological coherence: The identified metabolites clustered into biologically meaningful pathways rather than random patterns, supporting their validity.
  3. Multivariate validation: Clustring in PCA and PLS-DA analyses provides orthogonal support for the univariate findings.
  4. Consistency with literature: Key findings (e.g., oxidative stress markers, altered nucleotide metabolism) align with established PV pathophysiology.
  5. Technical reproducibility: Stringent QC filtering (%RSD <50%) and blank subtraction reduced technical false positives.

We view this study as hypothesis-generating rather than hypothesis-testing. The identified metabolic signatures require validation in independent cohorts with appropriate multiple testing corrections. We have added this important limitation to our revised manuscript

‘’Metabolite identification represents another important limitation; while a subset of metabolites was confirmed at MSI Level 1 using authentic standards and MS/MS fragmentation, many metabolites remain at Level 2 (putative annotation based on accurate mass and database matching). Although not all metabolites achieved Level 1 confirmation, the consistency of pathway enrichment patterns and the validity of relative quantification support the biological relevance of the observed metabolic differences between PV and SP. Overall, we view this study as hypothesis-generating rather than hypothesis-testing. The identified metabolic signatures require validation in independent cohorts with appropriate multiple testing corrections, and such confirmatory efforts will be essential before any clinical application.”

Comment 1.6: The role of deoxyuridine depletion as a biomarker is highlighted. Could this observation be confounded by dietary or pharmacological factors not reported in the study?

Response 1.6: We acknowledge that circulating nucleoside levels, including deoxyuridine, can in theory be influenced by dietary intake, vitamin supplementation, or pharmacological interventions such as antimetabolite therapy. To minimize these confounders, we excluded patients with conditions or comorbidities that might significantly alter nucleotide metabolism (e.g., renal or hepatic dysfunction, diabetes, or concurrent malignancy).

None of the patients included in this study were receiving cytotoxic, antimetabolite, or antiviral therapies that could directly affect pyrimidine metabolism.

While we did not collect detailed dietary intake data, all samples were obtained following an overnight fast, which reduces short-term dietary influence on plasma metabolite levels.

Moreover, the observed depletion of deoxyuridine was consistent across the PV cohort and aligned with mechanistic evidence of enhanced nucleoside salvage pathway utilization in malignant hematopoiesis, supporting its biological rather than purely dietary/pharmacologic origin.

Comment 1.7: The elevation of thyrotropin-releasing hormone in PV is unexpected. Were thyroid hormone levels measured in these patients to exclude subtle dysfunction?

Response 1.7: In routine follow-ups of PV patients, sT3, sT4, and TSH hormones are routinely monitored. No patient in the SP and PV groups was receiving thyroid hormone replacement therapy. All were in a euthyroid state

In accordance with routine clinical practice, thyroid function tests (serum TSH, free T3, and free T4) were evaluated during the follow-up of all PV patients. None of the patients in either the PV or SP groups were receiving thyroid hormone replacement therapy, and all demonstrated results within the reference ranges, consistent with a euthyroid state. Therefore, the observed elevation of thyrotropin-releasing hormone (TRH) in PV patients cannot be attributed to overt or subclinical thyroid dysfunction in this cohort.

We have also added information about this situation to the exclusion criteria to clarify that patients with thyroid dysfunction were not included in the study.

Comment 1.8:How do the findings on nicotinamide and B-vitamin metabolism relate to potential clinical interventions? Would the authors suggest vitamin supplementation studies, or is this premature?

Response 1.8:Our data reveal a significant dysregulation in the NAD+ biosynthetic pathway, a key aspect of B-vitamin metabolism, particularly in untreated PV patients. While these findings are compelling, we believe it is premature to advocate for broad vitamin supplementation studies. The complex interplay of metabolic pathways in PV means that untargeted supplementation could yield unpredictable effects on disease biology and treatment response. Therefore, rather than general nutritional interventions, we propose that the critical next step is focused mechanistic research. Such studies should aim to elucidate the precise role of the NAD+ pathway in PV pathophysiology and determine whether it constitutes a viable therapeutic target. This approach will enable the development of targeted inhibitors or specific dietary strategies, allowing this metabolic signature to be translated into a precise therapeutic intervention rather than a non-specific nutritional recommendation.

We have clarified this discussion in the revised manuscript.

“Our data point to a significant dysregulation in the NAD+ biosynthetic pathway, a key component of B-vitamin metabolism, particularly in untreated PV patients. However, we contend that it is premature to recommend supplementation trials; instead, future studies should focus on mechanistic research to determine if this pathway represents a viable therapeutic target.”

Reviewer 2 Report

Comments and Suggestions for Authors

The authors present an interesting and valuable study examining plasma metabolomic differences between primary and secondary polycythemia. By employing a range of multivariate statistical approaches—including principal component analysis, variable importance in projection analysis, heatmap analysis, pathway enrichment analysis, and cross-comparison analysis—they identify a set of distinct metabolites with potential as diagnostic biomarkers. These findings are novel, align well with previous research, and contribute meaningfully to the growing body of evidence in this area.

While the results are promising, the study is limited by the relatively small sample size and the absence of an independent validation cohort. In addition, the discussion section would benefit from being more concise and better structured to emphasize the most important biological insights. Addressing these points would further strengthen the clarity and overall impact of the manuscript.

Author Response

We thank the reviewer for their positive and constructive feedback on our manuscript. We are pleased that the reviewer found our study to be interesting and valuable, and that our findings are considered a meaningful contribution to the field. We agree with the reviewer's suggestions for improvement and have addressed them as follows:

Comment 2.1: Regarding the small sample size and absence of an independent validation cohort:

Response 2.1: The relatively small sample size, particularly in the secondary polycythemia (SP) group, resulted from stringent patient selection criteria and practical challenges. Our rigorous exclusion of confounding factors, such as specific comorbidities and steroid use, along with patient reluctance for extensive diagnostic evaluations, limited the cohort size. Despite these necessary reductions, our final sample size (40 PV and 25 SP patients) remains comparable to or larger than many published metabolomics studies in hematological disorders.

Comment 2.2: The discussion section would benefit from being more concise and better structured to emphasize the most important biological insights. Addressing these points would further strengthen the clarity and overall impact of the manuscript.

Response 2.2: We thank for your suggestion to make the discussion more concise and focused on the most important biological insights. We have revised the discussion section by removing some redundant sentences and restructuring the paragraphs to improve the flow and clarity. We believe these changes have strengthened the manuscript and better highlight the significance of our findings.

4.1.3

In PV, decreases in deoxyuridine (−2.71-fold) and inosine (−1.53-fold), although seemingly paradoxical for a proliferative disorder, align with a metabolic reprogramming strategy common in hematologic malignancies. However, mounting evidence from hematologic malignancy studies suggests that cancer cells frequently reprogram intracellular nucleotide metabolism to favor salvage pathway utilization over the de novo route.For example, Tran et al. (2024) demonstrated that circulating nucleosides, such as adenine and inosine, are major contributors to tumor purine pools and that blocking purine salvage impairs tumor growth in vivo [34].

In PV, the reduction in deoxyuridine (−2.71-fold) and inosine (−1.53-fold) appears paradoxical for a proliferative disease but reflects a metabolic reprogramming strategy seen in hematologic cancers. Evidence shows that malignant cells rewire their nucleotide metabolism to prefer the salvage pathway over de novo synthesis. For instance, Tran et al. (2024) demonstrated that circulating nucleosides like inosine are key to tumor purine pools, and inhibiting their salvage blocks tumor growth in vivo [34].

Similarly, broader reviews have emphasized that successful malignancies rely on upregulated nucleoside transport and salvage enzyme activity, streamlining the uptake and phosphorylation of extracellular nucleosides into deoxyribonucleotides [35,36].

Similarly, broader reviews emphasize that successful malignancies depend on upregulated nucleoside transport and salvage enzyme activity to streamline the conversion of extracellular nucleosides into deoxyribonucleotides [35,36].

From this perspective, the depleted extracellular levels of deoxyuridine and inosine in PV may reflect enhanced cellular uptake and elevated salvage flux into proliferating hematopoietic cells rather than a true shortage of nucleotide substrates. This model aligns well with the documented behaviors of malignant hematopoietic and myeloid cells, which often express high levels of nucleoside transporters (for example, ENT1/2 CNTs, CNTs) and salvage kinases, such as deoxycytidine kinase and thymidine kinase, to rapidly recycle circulating nucleosides into DNA precursors [37].

Thus, the low extracellular deoxyuridine and inosine in PV likely indicate increased cellular uptake and salvage activity in proliferating cells, rather than nucleotide substrate deficiency. This aligns with the known behavior of malignant hematopoietic and myeloid cells, which often overexpress nucleoside transporters (e.g., ENT1/2, CNTs) and salvage kinases to rapidly recycle nucleosides into DNA precursors [37].

4.1.6

Our metabolomic analysis revealed a substantial downregulation of 4-hydroxyretinoic acid in patients with PV compared to that in patients with SP (-2.62-fold, VIP: 1.52), providing novel insights into the disrupted retinoid signaling in PV pathogenesis. This finding supports the hypothesis that altered retinoid metabolism contributes to thrombocytosis and megakaryocytic hyperplasia, which are characteristic features of PV. Given that RA derivatives can inhibit JAK2, STAT3, and STAT5 phosphorylation [63], the depletion of 4-hydroxyretinoic acid in PV may represent a mechanism by which PV cells evade retinoid-mediated growth inhibition. The significant reduction in this bioactive retinoid metabolite could potentially explain the enhanced megakaryocytic proliferation in PV, as diminished retinoid signaling may impair the proper regulation of the GATA-1/FOG-1/NFE2 transcriptional network, which controls megakaryocyte maturation and platelet production [64].

Our metabolomic analysis identified a significant downregulation of 4-hydroxyretinoic acid in PV patients compared to SP (-2.62-fold), offering new insight into impaired retinoid signaling in PV. This supports the hypothesis that disrupted retinoid metabolism contributes to characteristic features like thrombocytosis and megakaryocytic hyperplasia. Since RA derivatives inhibit JAK2, STAT3, and STAT5 phosphorylation [63], the depletion of 4-hydroxyretinoic acid may allow PV cells to evade growth inhibition. This reduction could enhance megakaryocytic proliferation by impairing the regulation of the GATA-1/FOG-1/NFE2 transcriptional network, which controls megakaryocyte maturation and platelet production [64]

4.2.2

The marked downregulation of nicotinic acid adenine dinucleotide (NAAD) (-6.37-fold) and concurrent upregulation of nicotinamide ribotide (3.39-fold, VIP = 1.56) in patients not treated with cytoreductive therapy indicated a fundamental dysregulation of niacin (vitamin B3) metabolism and NAD+ biosynthetic pathways. The decreased NAAD levels suggest impaired de novo NAD+ synthesis via the kynurenine pathway, which is further supported by the significant downregulation of L-3-hydroxykynurenine (-3.25-fold, VIP = 1.45), a key intermediate in tryptophan-to-NAD+ conversion [73].

A significant downregulation of nicotinic acid adenine dinucleotide (NAAD) (-6.37-fold) and upregulation of nicotinamide ribotide (3.39-fold) in untreated PV patients indicated dysregulated niacin (vitamin B3) metabolism and NAD+ biosynthesis. Reduced NAAD suggests impaired de novo NAD+ synthesis through the kynurenine pathway, further supported by decreased L-3-hydroxykynurenine (-3.25-fold), a key intermediate in tryptophan-to-NAD+ conversion [73].

NAD+ serves as a critical cofactor in energy metabolism, including glycolysis, fatty acid oxidation, and the TCA cycle [74,75], and this metabolic disruption may contribute to the cellular energetic imbalance characteristic of MPNs. The reciprocal elevation of nicotinamide ribotide in patients treated with cytoreductive therapy likely represents a compensatory mechanism to maintain cellular NAD+ pools through enhanced salvage pathway activity. This metabolic adaptation suggests that patients not treated with cytoreductive therapy exhibit early-stage metabolic stress responses aimed at preserving NAD+-dependent cellular function.

NAD+ is a vital cofactor for energy metabolism (glycolysis, fatty acid oxidation, TCA cycle) [74,75], and its disruption may contribute to the energetic imbalance in MPNs. The elevated nicotinamide ribotide in treated patients likely reflects a compensatory salvage pathway activation to maintain NAD+ pools. This suggests untreated patients exhibit early metabolic stress responses to preserve NAD+-dependent functions.

Round 2

Reviewer 1 Report

Comments and Suggestions for Authors

Accept in its current form.